# Potassium-Deficient Nutrient Solution Affects the Yield, Morphology, and Tissue Mineral Elements for Hydroponic Baby Leaf Spinach (*Spinacia oleracea* L.)

**Christopher P. Levine and Neil S. Mattson \***

School of Integrative Plant Science, Cornell University, Ithaca, NY 14853, USA; cpl43@cornell.edu
\* Correspondence: nsm47@cornell.edu; Tel.: +1-607-255-0621

**Abstract:** Nutrient supply in hydroponics can significantly influence the nutrition, taste, texture, color, and other characteristics of fruit and vegetable crops. Chronic kidney disease (CKD) is a global health problem that frequently restricts a patient's consumption of high-potassium foods. CKD patients are advised to limit their consumption of many vegetables that are potassium (K)-rich. At the same time, reducing vegetable intake reduces the intake of healthy compounds such as vitamins, fibers and antioxidants, which are beneficial to CKD patients. In our study, we investigated the reduction of the K concentration in a hydroponic nutrient solution as a possible technique to decrease the K tissue concentration of baby leaf spinach, a dark green that is frequently recommended to be consumed in moderation for patients with CKD. A previously developed hydroponic fertilizer recipe that provides a platform to adjust individual nutrients was used to adjust K to 0, 10, 25, and 100% of the control K concentration. Tissue K levels were reduced by up to 91% with a consequent 61% reduction in dry weight and 76% reduction in fresh weight (yield) with respect to the control treatment. Overall, the results suggest that using a nutrient solution without K can significantly reduce K concentrations in baby spinach, although this will consequently reduce yields.

**Keywords:** hydroponics; deep water culture; potassium; low-potassium greens; chronic kidney disease; potassium restriction

## 1. Introduction

Hydroponics is a method for growing plants in a mineral nutrient solution without soil. A common form of hydroponics used for the commercial production of leafy greens and herbs is deep water culture, in which the plants roots are suspended in a nutrient-rich and oxygenated solution on a floating raft. Some benefits to the deep water culture hydroponics method over other hydroponic methods include the ability to maintain a stable water temperature and have crops survive during extended power outages [1].

Hydroponic systems are more water-efficient, providing the ability to grow crops more densely and allowing greater fertilizer use efficiency compared to conventional soil culture farming methods [2]. Furthermore, these systems can be implemented in areas with poor soil conditions and are suitable in urban areas where space is limited. In hydroponic systems, the nutrient solution can be adjusted and can significantly influence the nutrition, taste, texture, color, and other characteristics of fruit and vegetable crops.

In previous experiments that have investigated the effects of nutrients on hydroponic lettuce, one study found that the nitrate ($NO_3$) content in Samsunjokchukmyon lettuce can be decreased with the addition of potassium chloride and calcium chloride [3]. This finding is important because excessively high $NO_3$ food consumption can have negative health consequences for humans. Another study found that the iodine content in lettuce tissue can be increased (i.e., biofortification) without negatively affecting plant biomass, produce quality, or water uptake [4]. The supplementation of iodine, an essential trace element for humans, can alleviate the issue of insufficient iodine intake for a significant portion of the

world population. Overall, both of these studies successfully demonstrate efforts to modify the nutrient contents in hydroponic lettuce to obtain a desired health outcome.

Chronic kidney disease (CKD) is projected to increase up to 16.7% for Americans over the age of 30 by the year 2030 [5]. Furthermore, CKD is currently the 9th leading cause of death in the United States [6]. CKD patients have to limit their K intake by limiting the consumption of certain foods. The recommended K intake is generally restricted to less than 3 g per day for patients with reduced kidney function [7]. Furthermore, CKD patients with hyperkalemia are often instructed to reduce K-rich food intake, to cut K-rich food such as bananas and potatoes into smaller pieces, and to boil or soak food in a large volume of water prior to consumption. These food preparation procedures can result in the loss of other vitamins and minerals, as well the loss of desirable texture [8].

In standard hydroponic nutrient solutions, typical K levels frequently vary. The Hoagland and Arnon solution, a well-known hydroponic formula used for research, gives a K level of 234 ppm [9]. This recipe is not crop-specific or tailored for spinach; however, it has been widely used in hydroponic research for many years. Another hydroponic formula that is formulated specifically for leafy greens that was developed by well-established fertilizer companies recommends 235 ppm K [10]. The modified Sonneveld's solution, a hydroponic formula frequently used for hydroponic spinach research at Cornell University, recommends 210 ppm K [11].

In recent studies, K reduction experiments with hydroponic crops have been performed successfully with various types of lettuce, microgreens, melons, onions, and even spinach. One study was able to significantly decrease the K content of the outer leaves for green leaf, Boston, and Romaine lettuce varieties [8]. In a study investigating K reduction in microgreens, it was determined that a useful reduction of K can be achieved without negatively affecting the quality [12]. No significant differences in the microgreen study were observed in terms of shoot height, dry matter, proximate composition, or visual quality [12]. In a study investigating K reduction in hydroponic melon, it was determined that growing melons with 50% of the required potassium nitrate produced fruit with about 53% lower K compared to control [13]. In a study investigating K restriction in onions, it was determined that reducing K fertilization several weeks prior to harvest is a more effective method of K restriction rather than reducing K fertilization throughout the entire growing period [14]; however, darker greens such as spinach, which can be beneficial to CKD patients as well, should be studied further. A previous Japanese research report discussed reducing potassium in spinach for patients on dialysis [15]. Under their specific conditions, the potassium content was able to be reduced by as much as 79% relative to the control in their study [15]; however, this 79% reduction was due to the fact that not all potassium was eliminated in their treatment. In addition, light uniformity was not addressed.

Previous work with K restriction in hydroponic leafy greens has been primarily restricted to lettuce. Spinach is the second most popular leafy green consumed in the U.S.; however, information on the use of K restriction for spinach is not abundant in the United States. Similar research on low-potassium spinach has been performed in Italy as well [16]; however, only a 26.9% reduction in potassium relative to the control was achieved. This was due to the fact that not all potassium was removed in their study either. The purpose of our study is to determine how much K can be removed from baby spinach leaves by restricting the K input in the hydroponic fertilizer and the resulting impacts on plant yield and morphology. This also includes completely eliminating K and understanding the effects that this can have on spinach growth.

## 2. Materials and Methods

Our experiment was conducted in a glass greenhouse at Cornell University in Ithaca, New York (42° N and a longitude of 76° W). Trial 1 occurred between 5 November 2019 and 23 November 2019. Trial 2 occurred between 2 December 2019 and 21 December 2019. Trial 3 occurred between 27 January 2020 and 17 February 2020.

Prior to transplanting seedlings into the hydroponic treatment systems, 'Carmel (F1)' spinach seeds (Johnny's Selected Seeds, Winslow, Maine) were primed with Rootshield AG (BioWorks Inc., Victor, NY, USA) at a rate of 1 g per 1 oz of seed for Pythium inhibition. Rootshield powder and seeds were mixed together for 30 s in a plastic Ziplock bag prior to seed hydration. Spinach seeds and Rootshield AG were both kept refrigerated. After seed priming, 20 g of seeds were weighed and placed in a 400 mL graduated cylinder with 90.0 mL of reverse osmosis (RO) water. A magnetic stirring bar was inserted into the graduated cylinder and the beaker was covered and sealed with parafilm. The beaker was also covered in aluminum foil to keep the environment dark. The seeds were gently mixed using the stirring bar for 24 h at 22 °C.

Following the seed hydration procedures, hydrated spinach seeds were placed onto a moist paper towel in a sealed plastic container (34.6 cm × 21 cm × 12.4 cm). RO water was used to keep the paper towel moist. Then, 500 mL of RO water was poured into the bottom of the container. A plastic stand and a metal liner kept the moist towel elevated above the RO water to ensure the seeds were kept moist (but not suspended in water) for an additional 24 h at a constant temperature of 22 °C. The container was covered in aluminum foil to prevent light penetration.

Following the seed pregermination procedures, seeds with visible emerging radicles were sowed directly into 16-cell Styrofoam blocks (2.5 cm × 2.5 cm cell) with a custom-prepared soilless substrate (75% peat moss, 25% course perlite, 5.9 g·L$^{-1}$ limestone). The soilless substrate was mixed with RO water at a rate of 1000 mL water per kg of dry substrate. Following this procedure, the substrate was filtered through a 0.5 cm$^2$ density strainer to remove larger particles. Careful procedures were performed to ensure all Styrofoam blocks were covered with the soilless substrate at a uniform rate. After inserting the pregerminated seeds into the Styrofoam blocks (1 seed per cell), the blocks were covered in a plastic bag with aluminum foil and placed in an office room with a constant temperature of 22 °C for 3–4 days until the emergence of cotyledons.

Once the spinach plants had visible cotyledons, each Styrofoam block containing 16 cells was transferred into the 3.785 L bucket serving as a mini deep water culture hydroponic system. In our experiment, one 16-cell Styrofoam block (with 16 plants) in one 3.785 L bucket was considered an experimental unit. Our experiment involved 4 treatments, i.e., differing concentrations of K in the nutrient solution as described below, and each experimental run included 4 experimental units (buckets) for each treatment. Each hydroponic system was coated with silver spray paint to reduce light penetration into the nutrient solution and reduce algae growth. Each bucket also had one air stone that was on continuously throughout the experiment to supply dissolved oxygen. Our experiment was replicated over time for a total of three experimental runs. The experimental layout is shown in Figure 1 below.

With the emergence of cotyledons, Styrofoam blocks were floated on the hydroponic buckets containing the K-deficient treatments as described below to initiate the experimental conditions. The greenhouse bench used for the treatment period contained supplemental lighting from 2 high-pressure sodium lamps (LU400/H/ECO, General Electric, Boston, MA, USA). Supplemental light was supplied continuously (24 HR lighting) at 60 µmol·m$^{-2}$·s$^{-1}$, resulting in a supplemental daily light integral of 5.2 mol·m$^{-2}$·d$^{-1}$. Light levels (photosynthetic photon flux density, PPFD) were carefully mapped using a LI-COR LI-250A light meter (LICOR Biosciences, Lincoln, NE, USA) and buckets were randomly placed in a circular arrangement to ensure uniform light distribution between buckets. The ambient air temperature averaged 21 ± 1 °C during the day (between 0800 and 1700 HR) and 19 ± 1 °C during the night (between 1700 and 0800 HR). Temperature was monitored and adjusted using an Argus environmental control system (Argus Control Systems Ltd., Surrey, BC, Canada), in which data were collected every 2 s and averaged every 2 min. Nutrient solutions were replaced once after 1 week (168–172 h). The pH of each nutrient solution was monitored and adjusted as needed every 3–4 days to pH 5.5 ± 0.2.

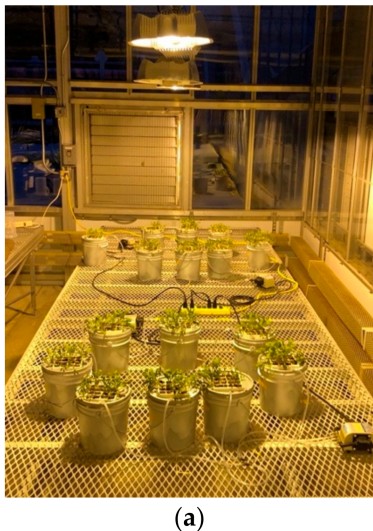
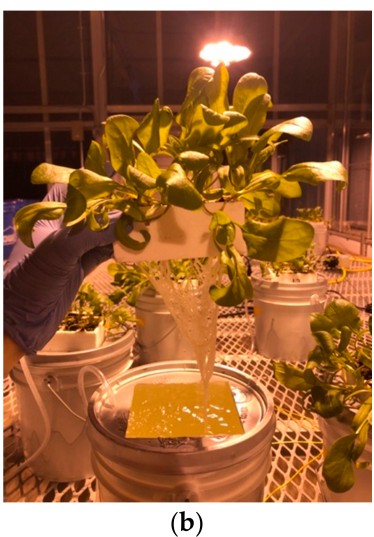

(**a**)    (**b**)

**Figure 1.** The experimental layout: (**a**) the 16 experimental units used in each run; (**b**) an individual bucket and the floating Styrofoam block that holds the spinach at the time of harvest.

The fertilizer treatments were established using a fertilizer recipe developed by Dr. Paul Nelson, North Carolina State University (pers. communication), which is similar to the recipe described by Taylor [17]. The control fertilizer solution consisted of 5 mM potassium nitrate ($KNO_3$), 5 mM calcium nitrate tetrahydrate ($Ca(NO_3)_2 \cdot H_2O$), 1 mM potassium phosphate monobasic ($KH_2PO_4$), 2 mM magnesium sulfate heptahydrate ($MgSO_4 \cdot 7H_2O$), 4 ppm Fe as Fe-diethylenetriaminepentaacetic acid (DTPA) (10% Fe), 9 μM manganese chloride tetrahydrate ($MnCl_2 \cdot 4H_2O$), 1.5 μM zinc chloride ($ZnCl_2$), 1.5 μM copper chloride dihydrate ($CuCl_2 \cdot 2H_2O$), 0.1 μM sodium molybdate dihydrate ($Na_2MoO_4 \cdot 2H_2O$), and 45 μM boric acid ($H_3BO_3$). The K reduction treatments were established by decreasing $KNO_3$ and $KH_2PO_4$ from the nutrient solution while increasing sodium nitrate ($NaNO_3$) and sodium phosphate monobasic ($NaH_2PO_4$) to reverse the losses in $NO_3$ and P. The resulting nutrient solution analysis for each treatment is shown in Table 1; thus, 4 K treatments were established: 0% (0 ppm K), 10% (19 ppm K), 25% (58 ppm K), and 100% (control, 244 ppm K). All other macro- and microelements except K and Na had similar values.

We decided to use the fertilizer recipe developed by Dr. Paul Nelson because it is widely used as a general hydroponic nutrient formula for nutrient-deficiency-related experiments at Cornell University and at other universities in the United States. We selected 0% K, 10% K, 25% K, and 100% K treatments based on Michaelis–Menten uptake kinetics [18]. We initially selected 0% and 100% concentrations, then based on Michaelis–Menten kinetics we chose the lower concentrations of 10% and 25% to increase the likelihood that we would find relevant results. Furthermore, we hoped to find a curve that would increase rapidly at low concentration and then exhibit slight increases at higher concentrations.

The nutrient solution preparation procedures were as follows: (1) a clean weighing boat and spatula must be obtained; (2) an appropriately sized beaker must be filled to about 80% of the final desired volume with Milli-Q ultra-pure water; (3) place the beaker on a magnetic stirrer and place a magnet in the beaker; (4) using an analytic scale, weigh out the desired chemical; (5) carefully place the chemical into the beaker; (6) using a lab washing bottle, squirt Milli-Q ultra-pure water onto the weighing boat to remove any additional chemicals and pour the liquid into the beaker; (7) wait until the stock solution is fully dissolved (this may take a few minutes); (8) pour the stock solution into a graduated cylinder; (9) fill the graduated cylinder with Milli-Q ultra-pure water until it is at its final desired volume; (10) using parafilm, seal the graduated cylinder and shake it so the stock solution is homogenously mixed; (11) pour the stock solution into a bottle; (12) update the

bottle label with the name and creation date (iron chelate stock solution must be wrapped in aluminum foil and placed in the refrigerator).

Prior to each trial, all hydroponic systems, Styrofoam blocks, tubing, and air stones were thoroughly cleaned and sterilized with a 10% diluted bleach solution to reduce the risk of developing Pythium or other diseases.

**Table 1.** Nutrient solution analysis of the 4 K treatment solutions (in ppm). Nutrient solution analysis was performed by JR Peters, Inc. (Allentown, PA, USA). Slight variations in mineral nutrient composition from theoretical to laboratory testing are expected and can be due to slight inconsistencies in preparation, as well as the fact that each commercial laboratory varies slightly in their methodology.

| Treatment | K | Ca | Mg | Na | P | S | Cl | NH4-N | NO3-N | Fe | Mn | B | Cu | Zn |
|---|---|---|---|---|---|---|---|---|---|---|---|---|---|---|
| 0% K | 0 | 169 | 41 | 122.75 | 28 | 58.2 | 3.57 | 1.22 | 182.62 | 3.96 | 0.46 | 0.44 | 0.1 | 0.1 |
| 10% K | 19 | 167 | 41 | 111.08 | 27 | 59.81 | 3.69 | 0.56 | 186.61 | 3.93 | 0.45 | 0.43 | 0.1 | 0.09 |
| 25% K | 58 | 166 | 41 | 92.05 | 27 | 58.17 | 3.66 | 0.43 | 181.68 | 3.79 | 0.53 | 0.42 | 0.1 | 0.1 |
| 100% K | 244 | 168 | 41 | 5.05 | 28 | 69.39 | 3.23 | 0.64 | 193.46 | 3.88 | 0.44 | 0.42 | 0.11 | 0.11 |

The K treatments were applied for 14 days (336–340 h), after which data were collected and 8 out of 16 spinach plants were randomly selected from each Styrofoam raft for measurements. We recorded the fresh weights (by cutting the spinach above the cotyledons) and determined leaf surface area values using a LI-3100C Area Meter (LI-COR Biosciences, Lincoln, NE, USA). Leaves were then placed in an oven at 70 °C to dry for 48 h to determine dry weight values. Within an experimental run, the leaves from each treatment were pooled together to form a bulk sample, which was then sent to a commercial lab (JR Peters Inc., Allentown, NJ, USA) for tissue analysis of the mineral element content. For the tissue analysis, tissue samples were dried for 24–72 h at 121 °C in a large convection oven. After the drying process, the tissue was ground into a fine powder then turned into ash in a muffle furnace at 450 °C for 16 h overnight. After this process, the ash was digested with 2 mL 6% $HNO_3$ acid. Following this, the solution was filtered through Whatman # 40 filter paper. The resulting solutions were analyzed for macronutrients (P, K, Ca, Mg) and micronutrients (Fe, Mn, B, Cu, Zn, Mo, Al, Na) using an inductively coupled plasma atomic emission spectrometer. Nitrogen was measured via combustion analysis using a Thermo Scientific Flash Analyzer (Dr. Cari Peters, pers. communication).

The experiment followed a completely randomized design, with four experimental units for each of the 4 K treatments. The experiment was repeated over time for a total of three experimental runs. Data were analyzed using JMP Pro 14 software, a statistical analysis software developed by the JMP business unit of the SAS Institute, using a Tukey–Kramer honest significant difference (HSD) test at $\alpha = 0.05$ to determine significant differences among measured parameters based on K treatment.

## 3. Results

Spinach plants with the 0% K treatment exhibited the lowest K concentration in tissue of 0.8%, while the 25% K and 100% K treatments showed statistically the same K tissue concentrations of 8.7–9.1% (Figure 2a). Spinach plants exhibited a 61% reduction in dry weight (DW) in the 0% K treatment as compared to the control (100% K) (Figure 2b). The DW values of the 10% K and 25% K plants were similar to each other but significantly smaller than the control (100% K). The spinach plants receiving the 0% K treatment showed a 76% reduction in fresh weight (yield) with respect to the control (Figure 2c). The fresh weight (FW) for 25% K treatment was not significantly smaller than for control plants, whereas the 10% K treatment was significantly smaller than control and 25% K treatments but larger than the 0% K treatment. The leaf surface area for the 25% K treatment was not significantly smaller than for the control plants, whereas the 10% K treatment was significantly smaller than the control and 25% K treatments (Figure 2d). The 0% K treatment showed a significantly lower leaf surface area compared to the 10% K treatment. Regarding

the specific leaf area, the 100% K and 25% K treatments had the highest specific leaf area values, while the 0% K treatment had the lowest specific leaf area (Figure 2e).

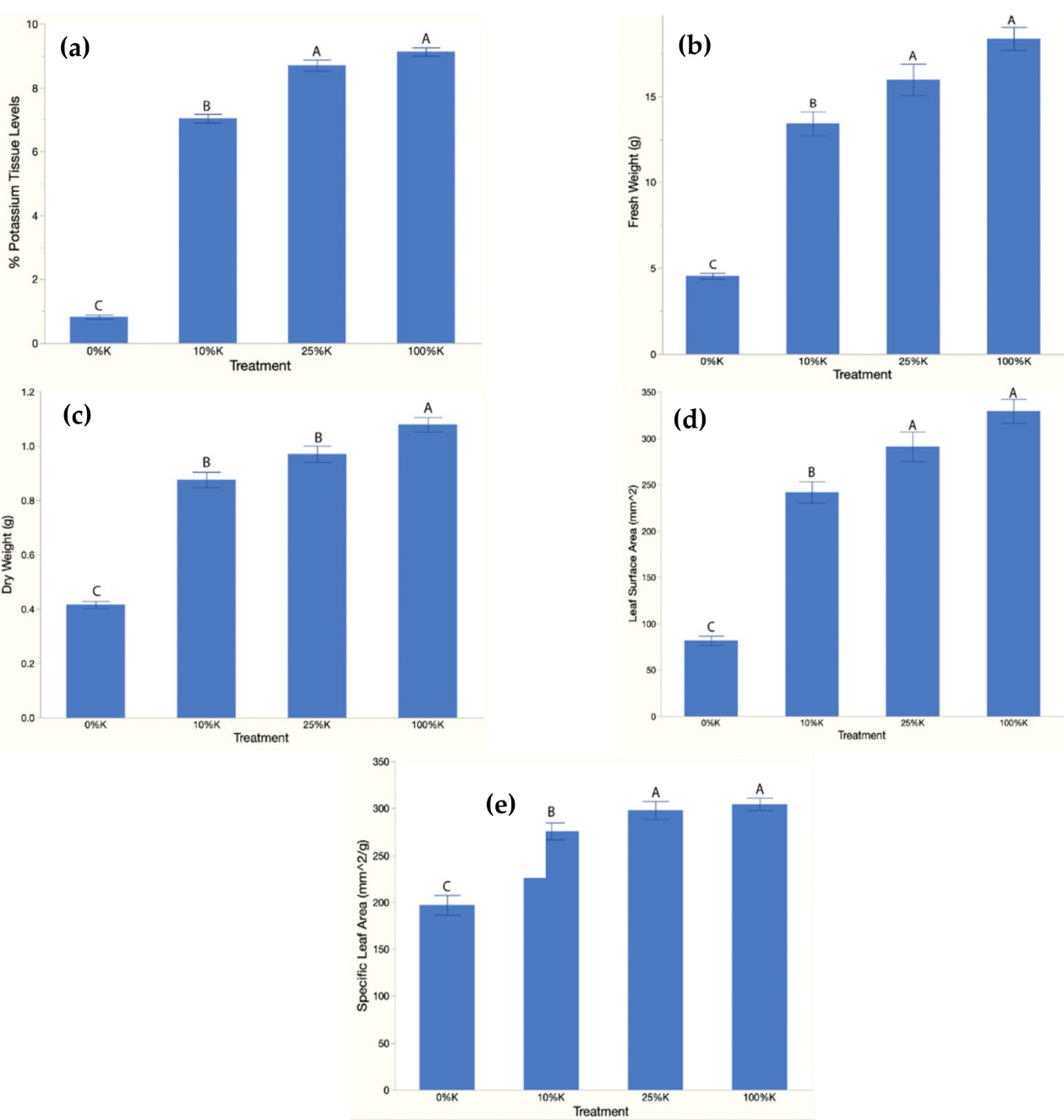

**Figure 2.** Effects of potassium-deficient hydroponic nutrient solutions (0%, 10%, 25%, and 100% of control K concentration) on different parameters: (**a**) %K tissue levels; (**b**) spinach plant fresh weight; (**c**) spinach plant dry weight; (**d**) leaf surface area; (**e**) specific leaf area. Bars represent means (± standard error) of 12 experimental units (across 3 experimental runs) for each treatment. Different letters among means indicate significant differences using the Tukey–Kramer HSD test at $\alpha$ = 0.05.

Representative mature leaves and plants from each treatment at harvest are shown in Figure 3a,b, respectively. Regarding the tissue analysis of other nutrients beyond K, we noticed increases in Mn, Mg, and B absorption as the K treatment decreased (Table 2). We also noticed a significant increase in Na absorption as K treatments decreased; however, the Na increase was expected, since the replacement nutrients to balance out the lost P and $NO_3$ had to be replaced with chemicals that were derived from Na.

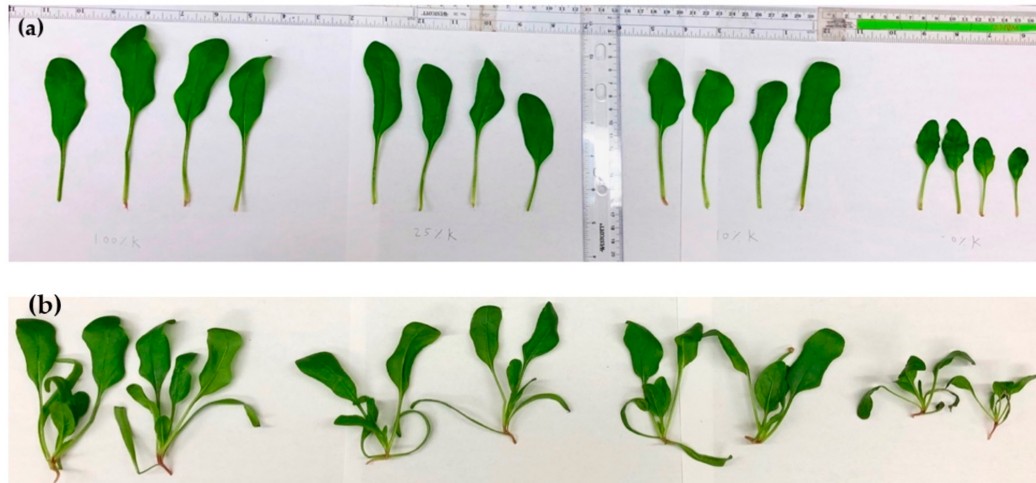

**Figure 3.** Images of randomly selected samples of harvested spinach with K fertilizer treatment samples decreasing from left to right (100% K, 25% K, 10% K, 0% K): (**a**) the 4 oldest spinach leaves; (**b**) 2 whole spinach plants from each treatment cut at the base of the hypocotyl.

**Table 2.** Average plant tissue analysis for the 4 potassium treatments. Plant tissue analysis was performed on a dry weight basis for shoot tissue by JR Peters, Inc. (Allentown, PA, USA).

| | Percent (%) | | | | | | mg/kg (ppm) | | | | | | | |
|---|---|---|---|---|---|---|---|---|---|---|---|---|---|---|
| Treatment | N | P | K | Ca | Mg | S | Fe | Mn | B | Cu | Zn | Mo | Na | Al |
| 0% K | 6.4 | 1.1 | 0.8 | 0.9 | 1.5 | 0.18 | 105.3 | 136.3 | 48.3 | 10.7 | 63.8 | 5.3 | 38,697 | 16.2 |
| 10% K | 6.1 | 0.8 | 7.0 | 0.9 | 1.3 | 0.12 | 85.1 | 95.6 | 35.4 | 7.2 | 50.0 | 3.6 | 14,758 | 6.9 |
| 25% K | 5.4 | 0.8 | 8.7 | 0.8 | 1.0 | 0.13 | 75.3 | 79.8 | 32.3 | 6.3 | 48.6 | 4.6 | 5673 | 9.1 |
| 100% K | 5.7 | 0.8 | 9.1 | 0.6 | 0.9 | 0.14 | 76.6 | 76.1 | 32.1 | 7.1 | 48.3 | 2.6 | 471 | 9.7 |

## 4. Discussion

Overall, spinach plants grown in 0% K nutrient solutions produced significantly lower tissue K concentrations than control (100% K), while increasing the Na content. Consequently, there were also reductions in fresh and dry weight yields as the potassium concentration decreased. This suggests that K reduction in spinach plants is possible but requires further investigation to find the optimal parameters needed to maximize K restriction while minimizing yield loss. For example, we found K concentration could be reduced by 91% with a corresponding FW yield decrease of 76% when comparing 0% K treatment to 100% K treatment. It is also important to mention here that the 0% K treatment still did have a small concentration of K in the tissue. We suspect that most of the potassium from the 0% K treatment must have come from the seeds. It may be possible that some K was contained in the custom media substrate mix because laboratory-grade substrates were not used; however, the latter is less likely because our selected substrates did not contain a significant source of K. Overall, the average dry weight yield per bucket (16 spinach plants) was approximately 0.83 g for the 0% K treatment, with about 6.64 mg of K in total for all 16 plants. To put this in perspective, 6.64 mg is about 0.3% of the potassium intake for a CKD patient placed on a 2000 mg per day potassium-restricted diet (a potassium-restricted diet is typically about 2000 mg per day); however, a physician or dietitian will advise each patient as to the specific level of restriction they need based on their individual health (Dr. Michelle Lubetzky, pers. communication). The control treatment (100% K) dry weight yield per bucket was approximately 2.15 g, with about 195.65 mg of K in total for all 16 plants. To put this in perspective, 195.65 mg is about the same weight as almost 2 honeybees. Since low-K spinach may be considered a specialty crop, the reduction in yields may still be justified, since the crop could be provided at a premium to CKD patients. Based on

three repeated trials, spinach was able to consistently be grown and harvested despite the significant reductions in yield.

In the past, studies on potassium reduction in spinach have been performed. In one study, spinach was reported to be reduced by as much as 79% relative to the control [15]. In another study, a 26.9% reduction of K in spinach was achieved relative to the control [16]. Neither of these prior studies included a 0% K treatment across the entire duration of the growth period. Quantifying the maximum amount of potassium that can be removed under experimental conditions is important because this will provide a baseline for researchers to use to understand how their unique potassium restriction methods compare when 0% K is included as a growing treatment.

In addition, the question arises of how to implement these practices in a commercial farm setting. Greenhouse hydroponic growers would have to adhere to strict protocols in order to reduce Pythium root rot issues if deciding to grow hydroponic spinach. Hydroponic spinach is highly susceptible to Pythium root rot if proper sanitation and other cultural strategies are not implemented. Secondly, close attention to the nutrient solution thresholds must be paid and random tissue analysis samples should be routinely collected to ensure K is being properly reduced for CKD patients. Since the consequences of consuming high-K spinach are likely severe for CKD patients, it is necessary to ensure that the spinach is produced in a highly controlled and regulated manner. Furthermore, more inexpensive fertilizer chemicals should be considered if this technique is to be implemented in a commercial setting. The fertilizer chemicals used in our research study are likely impractical for commercial growers because of the high purity and cost of the chemicals, as well as the high Na content in our alternative chemicals. More research should go into deciding which fertilizer chemicals are ideal substitutes to eliminate K while at the same time not reducing the other essential element thresholds. Lastly, since our research was conducted in a research greenhouse, stringent sanitation was implemented. In a commercial farm setting, low-K spinach may be more susceptible to unwanted pests and other diseases, since the spinach is likely in a stressed state of suboptimal nutrient conditions; this should be considered as well.

In future studies, it may also be worth investigating whether starting spinach growth with standard hydroponic fertilizers and removing K from the nutrient solution later on in the growth cycle may be more effective at removing K from the spinach tissue. A study on removing the K content from melons beginning four weeks into the growth cycle led to successful results in a Japanese study [13]. Furthermore, in another study focused on removing nitrate from aeroponically grown lettuce, the authors suggested that adjusting the fertilizer on a day and night basis can be effective in reducing nitrate levels [19]. In addition, alternative chemicals that do not contain Na should be considered. Utilizing potassium sulfate instead of potassium chloride is one example. Excess Na consumption beyond 2000 mg can have negative effects on diuretic hypertension therapy response [17]; therefore, alternative Na-free chemicals should be investigated further and considered in future trials.

Overall, there may be other strategies and growing techniques that can more effectively reduce K without significantly reducing yields with respect to standard fertilizer treatment. There are also likely alternative chemicals that would not increase Na levels. More research should be done to investigate other options to achieve the desired K content in spinach.

**Author Contributions:** C.P.L. led the experimental design, execution, and analysis. N.S.M. contributed to the experimental design and interpretation of results. C.P.L. wrote the paper, with editing contributions from N.S.M. Both authors have read and agreed to the published version of the manuscript.

**Funding:** This research was funded by the Cornell University College of Agriculture and Life Sciences (CALS) Charitable Trust. Any opinions, findings, conclusions, or recommendations expressed in this publication are those of the authors and do not necessarily reflect the view of the CALS Charitable Trust.

**Institutional Review Board Statement:** Not applicable.

**Informed Consent Statement:** Not applicable.

**Data Availability Statement:** The data that supports our paper can be confirmed by contacting JR Peters, Inc. (Allentown, PA, USA). Solution analysis Lab I.D. includes: 20-292241, 20-292242, 20-292243, 20-292244. Plant tissue analysis Lab I.D. includes: 19-290249, 20-290994, 20-290991, 20-290992, 20-290993, 20-292951, 20-292954, 20-292953, 20-292952, 19-290247, 19-290246, 19-290248.

**Acknowledgments:** We would like to thank Francoise Vermeylen from the Cornell Statistical Consulting Unit for her assistance and guidance with the JMP analysis. We would also like to thank Kendra Hutchins from the Cornell Controlled Environment Agriculture Lab for helping acquire the needed supplies for the experiment and sending the spinach samples for tissue analysis.

**Conflicts of Interest:** The authors declare no conflict of interest.

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
