# Peer review of "Potassium-Deficient Nutrient Solution Affects the Yield, Morphology, and Tissue Mineral Elements for Hydroponic Baby Leaf Spinach (Spinacia oleracea L.)"

_horticulturae, doi:10.3390/horticulturae7080213_

Round 1

Reviewer 1 Report

The article fits into the Journal's scope. Subject matter is original and important. All research components are present and clearly stated. The text is adequately written, although very few editorial and language errors should be corrected – but typographical and linguistic errors can be easily fixed. The abstract is well‐written, concise and clear. Introduction is well‐written and concise. Objective of the study is clearly presented. Results are presented logically. Conclusions are drawn from the analysis of the data collected. References are adequate and based on relevant literature.

From my point of view description of analytical procedures regarding minerals determination should be added. Also quality of the given results can be improved by addition of given results standard deviations.

Author Response

From my point of view description of analytical procedures regarding minerals determination should be added:

Response:  The following explanation has been added to the materials and methods section:

The nutrient solution preparation procedures are as followed: First, a clean weigh boat and a spatula must be obtained. Second, an appropriately sized beaker must be filled to about 80% of the final desired volume with Milli-Q ultra-pure water. Third, place the beaker on a magnetic stirrer and place a magnet in the beaker. Four, using an analytic scale, weight out the desired chemical. Fifth, carefully place the chemical into the beaker. Six, using a lab wash bottle, squirt Milli-Q ultra-pure water onto the weigh boat to remove any additional chemicals and pour into beaker. Seven, wait until the stock solution is fully dissolved (may take a few minutes). Eight, pour the stock solution into a graduated cylinder. Nine, fill the graduated cylinder with Milli-Q ultra-pure water until it is at its final desired volume. Ten, using parafilm, seal the graduated cylinder and shake so the stock solution is homogenously mixed. Eleven, pour the stock solution into a bottle. Twelve, update bottle label with the name and date created. (Iron chelate stock solution must be wrapped in aluminum foil and placed in the refrigerator.

Also, quality of the given results can be improved by addition of given results standard deviations.

Response: I included standard error bars in all of the graphs under the results section.  

Reviewer 2 Report

The manuscript “Low Nutrient Solution Potassium Affects Yield, Morphology, and Tissue Mineral Elements for Hydroponic Baby Leaf Spinach (Spinacia oleracea L.)” reports the effects of potassium in the nutrient solution on production and tissue mineral elements of baby leaf spinach.
The subject of the study falls within the general scope of Horticulturae.

Title
I suggest changing the title. It does not give the idea of the content of the work because there is a 0% K treatment.
Introduction
The introduction is very generic and refers to soilless crops, a technique widely used for the production of leafy vegetables. What are the typical levels of K in solution for the hydroponic production of baby leaf spinach on the market? The reduction of K in solution affecting a reduction of K in plant tissue is not new at all: Review lines 72-77, also in the light of the reported works:
Ogawa, A.; Eguchi, T.; Toyofuku, K. Cultivation methods for leafy vegetables and tomatoes with low potassium content for dialysis patients. Environ. Control Biol. 2012, 50, 407–414.
D’Imperio, M.; Montesano F.F.; Renna M.; Parente A.; Logrieco A.F.; Serio F. Hydroponic production of reduced-potassium swiss chard and spinach: a feasible agronomic approach to tailoring vegetables for chronic kidney disease patients. Agronomy 2019, 9, 627.

Materials and Methods.
Lines 137-146: ….Fertilizer solution consisted of 5 mM potassium nitrate (KNO3), …., 1 mM potassium phosphate monobasic (KH2PO4),
Then:
atomic weight of potassium: 39,1 (39,0983).
K ppm in the nutrient solution: 39,1 x 6 mM: 235 (234,6) ppm. Why 244 ppm of K (tab.1)?
About Ca:
Lines 137-146: ….Fertilizer solution consisted of….. 5 mM calcium nitrate tetrahydrate (Ca(NO3)2·H2O
Then:
atomic weight of calcium: 40,1.
Ca ppm in the nutrient solution: 40,1 x 5 mM: 200 (200,5) ppm. Why 166-169 ppm of Ca (tab.1)?

and so on for other macronutrient...

The potassium was replaced with sodium (lines 142-144) and this may have affected the results. In future experiments it would be appropriate to evaluate alternative strategies (eg. Potassium sulphate?), Especially if potassium is not completely excluded from the nutrient solution (which is quite impractical from a commercial point of view ...)

Line 162: describe briefly methods used for tissue analysis of mineral element content.
Line 166, JMP: ?

Results and discussion:
Line 215: Plant tissue analysis on fresh weight or dry weight?
Tips for broadening the discussion: In absolute terms, how much potassium was removed from the plants in one pot? Where does this potassium in the K0% treatment come from?
Lines 235-238. What are the other potential cultivation strategies and techniques that can more effectively reduce K without significantly reducing yields compared to a standard fertilizer treatment?
Can excess sodium have a negative effect on CKD patients?

Reference:
A thorough study is needed on the issue of reducing potassium in vegetables.

Although the subject of the study falls within the general scope of "Horticulturae", the manuscript needs to be greatly improved.

Author Response

Title
I suggest changing the title. It does not give the idea of the content of the work because there is a 0% K treatment.

Response: The title has been changed to the following:

Potassium Deficient Nutrient Solution Affects Yield, Morphology, and Tissue Mineral Elements for Hydroponic Baby Leaf Spinach (Spinacia oleracea L.)”

Introduction
The introduction is very generic and refers to soilless crops, a technique widely used for the production of leafy vegetables. What are the typical levels of K in solution for the hydroponic production of baby leaf spinach on the market?

Response: I added the following to the Introduction section:

In standard hydroponic nutrient solutions, typical K levels frequently vary. The Hoagland & Arnon Solution, a well-known hydroponic formula for research has 234 ppm K [9]. This recipe is not crop specific or tailored for spinach. However, it has been widely used in hydroponic research for many years. Another hydroponic formula that is formulated specifically for leafy greens that was developed by well-established fertilizer companies recommends 235 ppm K [10]. A modified Sonneveld’s Solution, a hydroponic formula frequently used for hydroponic spinach research at Cornell University has 210 ppm K [11].

The reduction of K in solution affecting a reduction of K in plant tissue is not new at all: Review lines 72-77, also in the light of the reported works:

Ogawa, A.; Eguchi, T.; Toyofuku, K. Cultivation methods for leafy vegetables and tomatoes with low potassium content for dialysis patients. Environ. Control Biol. 2012, 50, 407–414. 
D’Imperio, M.; Montesano F.F.; Renna M.; Parente A.; Logrieco A.F.; Serio F. Hydroponic production of reduced-potassium swiss chard and spinach: a feasible agronomic approach to tailoring vegetables for chronic kidney disease patients. Agronomy 2019, 9, 627.

Response: I revised the section of the paragraph and also cited one of the articles that you mentioned above.

Previous work with K restriction in hydroponic leafy greens has been primarily restricted to lettuce. Spinach is the second most popular leafy green consumed in the U.S. However, previous information on the use of K restriction for spinach is limited in the United States. Similar research on low potassium spinach has been performed in Italy [15]. The purpose of this study is to determine how much K can be removed from baby spinach leaves by restricting K input in the hydroponic fertilizer and the resulting impact on plant yield and morphology.

Materials and Methods.
Lines 137-146: ….Fertilizer solution consisted of 5 mM potassium nitrate (KNO3), …., 1 mM potassium phosphate monobasic (KH2PO4),
Then:
atomic weight of potassium: 39,1 (39,0983).
K ppm in the nutrient solution: 39,1 x 6 mM: 235 (234,6) ppm. Why 244 ppm of K (tab.1)?

Response: The numbers in lines 137-146 were theoretical calculations. Table 1 were the actual values tested with ICP analysis. I included the following disclosure in Table 1:

*slight variation in mineral nutrient composition from theoretical to laboratory testing is expected and can be due to slight inconsistencies in preparation as well as each commercial laboratory varies slightly in their methodology

About Ca:
Lines 137-146: ….Fertilizer solution consisted of….. 5 mM calcium nitrate tetrahydrate (Ca(NO3)2·H2O 
Then:
atomic weight of calcium: 40,1.
Ca ppm in the nutrient solution: 40,1 x 5 mM: 200 (200,5) ppm. Why 166-169 ppm of Ca (tab.1)?

Response: The numbers in lines 137-146 were theoretical calculations. Table 1 were the actual values tested with ICP analysis. I included the following disclosure in Table 1:

*slight variation in mineral nutrient composition from theoretical to laboratory testing is expected and can be due to slight inconsistencies in preparation as well as each commercial laboratory varies slightly in their methodology

and so on for other macronutrient...

The potassium was replaced with sodium (lines 142-144) and this may have affected the results. In future experiments it would be appropriate to evaluate alternative strategies (eg. Potassium sulphate?), Especially if potassium is not completely excluded from the nutrient solution (which is quite impractical from a commercial point of view ...)

Response: I will include the alternative strategies of using potassium sulfate in my discussion section. Furthermore, I will mention that my results are under experimental conditions and that my specific nutrient strategy may not be practical from a commercial production standpoint.

Text inserted in lines 318-324: Furthermore, more inexpensive fertilizer chemicals should be considered if this technique were to be implemented in a commercial setting. The fertilizer chemicals used in our research study are likely impractical for commercial growers because of the high purity and cost of the chemicals as well as the high Na content in our alternative chemicals. More research should go into deciding what fertilizer chemicals are ideal substitutes to eliminate K while at the same time not reducing the other essential element optimal thresholds.

Text inserted in lines 335-338:

In addition, alternative chemicals that do not contain Na should be considered. Potassium sulfate is one example instead of utilizing potassium chloride. Excess Na consumption beyond 2000 mg can have negative effect on hypertension diuretic therapy response (17). Therefore, alternative Na free chemicals should be investigated further and considered in future trials.

Line 162: describe briefly methods used for tissue analysis of mineral element content.

Response: Following text has been inserted.

For the tissue analysis, tissue samples were dried for 24-72 hours at 121º C in a large convection oven. Following the drying process, the tissue was ground into a fine powder then ashed in a muffle furnace at 450° C for 16 hours overnight. The ash was digested with 2 mL 6% HNO3 acid and the solution was filtered through Whatman # 40 filter paper. The resulting solutions were analyzed for macronutrients (P, K, Ca, Mg) and micronutrients (Fe, Mn, B, Cu, Zn, Mo, Al, Na) by Inductively Coupled Plasma Atomic Emission Spectrometer. Nitrogen was measured by combustion analysis by a Thermo Scientific Flash Analyzer (pers. communication with Dr. Cari Peters).   

Line 166, JMP: ?

Response: I included a short explanation of what JMP is below and it is inserted at the end of the materials and methods section.

“Data was analyzed using JMP Pro 14 software, a statistical analysis software developed by the JMP business unit of the SAS Institute, using a Tukey-Kramer HSD test to determine significant differences in measured parameters based on K treatment.”

Results and discussion:
Line 215: Plant tissue analysis on fresh weight or dry weight?

Response:   Tissue analysis was performed on dry weight tissue. I made a revision in the manuscript to mention this detail.

Tips for broadening the discussion: In absolute terms, how much potassium was removed from the plants in one pot? Where does this potassium in the K0% treatment come from?

Response: I have the following text inserted in my discussion section:

What is also important to mention here is that the 0%K treatment still did have a small concentration of K in the tissue. We suspect most of the potassium from 0%K treatment must come from the seed. It may be possible that some K was in the custom media substrate mix because laboratory grade substrates were not used. However, the latter is more unlikely because our selected substrates do not contain a significant source of K. Overall, the average dry weight yield per bucket (16 spinach plants) was approximately 0.83 grams for the 0%K treatment with about 6.64 mg of K in total with all 16 plants. To put this in perspective to something relatable, 6.64 mg is about the same weight as 2 to 3 mosquitoes combined. The control treatment (100%K) dry weight yield per bucket was approximately 2.15 grams with about 195.65 mg of K in total with all 16 plants. To put this in perspective to something relatable, 195.65 mg is about the same weight as almost 2 honeybees.

Lines 235-238. What are the other potential cultivation strategies and techniques that can more effectively reduce K without significantly reducing yields compared to a standard fertilizer treatment?

Response: Please see line 329-335. I have mentioned two other possible techniques that can be more effective at reducing K levels in spinach. Changing nutrient solutions on a day/night basis is one method. The other method included restricting potassium in nutrient solution a few weeks prior to harvest.

Can excess sodium have a negative effect on CKD patients?

Response: Yes, excess Na consumption beyond 2000 mg can have negative effect on hypertension Diuretic therapy response. I have the following text inserted in the article:

In addition, alternative chemicals that do not contain Na should be considered. Potassium sulfate is one example instead of utilizing potassium chloride. Excess Na consumption beyond 2000 mg can have negative effect on hypertension diuretic therapy response (17). Therefore, alternative Na free chemicals should be investigated further and considered in future trials.

Reference
A thorough study is needed on the issue of reducing potassium in vegetables.

Response: Will work on and add additional sources.

Although the subject of the study falls within the general scope of "Horticulturae", the manuscript needs to be greatly improved.

Reviewer 3 Report

General Comments

Dear authors

The manuscript entitled “Low nutrient solution potassium affects Yield morphology, and tissue mineral elements for Hydroponic Baby leaf Spinach (Spinacia oleracea L.)”

It is clear from the above manuscript that the authors must ensure that the manuscript adheres to the author's guidelines outlined in the “instructions for authors on the journal website since is not followed the same format throughout the manuscript.

Specific Comments

The results presented in the manuscript are interesting and well-structured and provides some good initial research results on how we can monitor the nutrient added in the hydroponic unit. This is an interesting paper that provides production information along with hydroponics systems.  

On the methodology please explain briefly why you chose these four concentrations of potassium?

In the results figure 2 illustrate only the fresh and dry weight of the baby spinach, but not the biomass of the baby spinach. The most important variable is the biomass and not necessarily the leaf surface area or area of branches and leafs. The biomass gain determines the amount of nutrients retained in the biomass.

The discussion needs to be improved both in content and organization, analyzing the results of the study in a more comprehensive way, because It's unclear what this manuscript wants to discuss.

Furthermore, I am not sure that the conclusions are novel, in the sense that the effect of low nutrient solution potassium further, affect the physiological response of the species Spinacia oleracea ;

Also, the question arises how this result translates into practice?

Author Response

Specific Comments

The results presented in the manuscript are interesting and well-structured and provides some good initial research results on how we can monitor the nutrient added in the hydroponic unit. This is an interesting paper that provides production information along with hydroponics systems.  

 On the methodology please explain briefly why you chose these four concentrations of potassium?

Response: We decided to use the fertilizer recipe developed by Dr. Paul Nelson because it is widely used as a general hydroponic nutrient formula for nutrient deficiency related experiments at Cornell University and at other universities in the United States. We selected 0%, 10%, 25%, and 100% K treatments based on Michaelis-Menten uptake kinetics [17]. We wanted 0% and 100% concentrations and then based on Michaelis-Menten kinetics choose the lower concentrations of 10% and 25% to increase the likelihood that we would find relevant results. Furthermore, we hoped to find a curve that increases rapidly at low concentration and then exhibits little increase at higher concentrations.

In the results figure 2 illustrate only the fresh and dry weight of the baby spinach, but not the biomass of the baby spinach. The most important variable is the biomass and not necessarily the leaf surface area or area of branches and leafs. The biomass gain determines the amount of nutrients retained in the biomass.

Response: We collected the fresh and dry weight (biomass) of the entire plant above the cotyledons. We would prefer to keep the leaf surface area because it can provide good insight for the commercial production of this spinach, insight on commercial transport, and consumer experience with ideal leaf surface areas.

The discussion needs to be improved both in content and organization, analyzing the results of the study in a more comprehensive way, because It's unclear what this manuscript wants to discuss.

Response: Please see the revised discussion section.

Furthermore, I am not sure that the conclusions are novel, in the sense that the effect of low nutrient solution potassium further, affect the physiological response of the species Spinacia oleracea ;

            Response: I revised my manuscript and indicated that this is not new or novel research.

Also, the question arises how this result translates into practice?

Response: I added the following text into the discussion section on how to translate this research into practice.

In addition, the question arises on how to implement these practices in a commercial farm setting. Greenhouse hydroponic growers would have to adhere to strict protocols in order to reduce Pythium root rot issues if deciding to grow hydroponic spinach. Hydroponic spinach is highly susceptible to Pythium root rot if proper sanitation and other cultural strategies are not implemented. Secondly, close attention to the nutrient solution thresholds must be carefully monitored and random tissue analysis samples should be routinely collected to ensure K is being properly reduced for CKD patients. Since the consequences of consuming high K spinach is likely severe for CKD patients, it is necessary to ensure the spinach is produced in a highly controlled and regulated manner. Furthermore, more inexpensive fertilizer chemicals should be considered if this technique were to be implemented in a commercial setting. The fertilizer chemicals used in our research study are likely impractical for commercial growers because of the high purity and cost of the chemicals as well as the high Na content in our alternative chemicals. More research should go into deciding what fertilizer chemicals are ideal substitutes to eliminate K while at the same time not reducing the other essential element optimal thresholds. Lastly, since our research was conducted in a research greenhouse, stringent sanitation was implemented. In a commercial farm setting, low K spinach may be more susceptible to unwanted pests and other diseases since the spinach is likely in a stressed state of suboptimal nutrient conditions. This should be considered as well.

Reviewer 4 Report

A continuation of the experiment is recommended for Ca and nitrate content/concentration which are highly influencing the kidney function.

Author Response

A continuation of the experiment is recommended for Ca and nitrate content/concentration which are highly influencing the kidney function.

Response: I will take this into consideration for follow up experiments in the future.
